# From Vaccine Vector to Oncomodulation: Understanding the Complex Interplay between CMV and Cancer

**DOI:** 10.3390/vaccines7030062

**Published:** 2019-07-09

**Authors:** Nicole A. Wilski, Christopher M. Snyder

**Affiliations:** Department of Microbiology and Immunology, Sidney Kimmel Cancer Center, Thomas Jefferson University, Philadelphia, PA 19107, USA

**Keywords:** cytomegalovirus, cancer vaccines, oncomodulation, anti-tumor immunity

## Abstract

Cytomegalovirus (CMV) is a herpesvirus that establishes a persistent, but generally asymptomatic, infection in most people in the world. However, CMV drives and sustains extremely large numbers of antigen-specific T cells and is, therefore, emerging as an exciting platform for vaccines against infectious diseases and cancer. Indeed, pre-clinical data strongly suggest that CMV-based vaccines can sustain protective CD8^+^ T cell and antibody responses. In the context of vaccines for infectious diseases, substantial pre-clinical studies have elucidated the efficacy and protective mechanisms of CMV-based vaccines, including in non-human primate models of various infections. In the context of cancer vaccines, however, much less is known and only very early studies in mice have been conducted. To develop CMV-based cancer vaccines further, it will be critical to better understand the complex interaction of CMV and cancer. An array of evidence suggests that naturally-acquired human (H)CMV can be detected in cancers, and it has been proposed that HCMV may promote tumor growth. This would obviously be a concern for any therapeutic cancer vaccines. In experimental models, CMV has been shown to play both positive and negative roles in tumor progression, depending on the model studied. However, the mechanisms are still largely unknown. Thus, more studies assessing the interaction of CMV with the tumor microenvironment are needed. This review will summarize the existing literature and major open questions about CMV-based vaccines for cancer, and discuss our hypothesis that the balance between pro-tumor and anti-tumor effects driven by CMV depends on the location and the activity of the virus in the lesion.

## 1. Introduction

Cancer immunotherapy has become a key player in cancer treatment as an alternative to chemotherapy and radiation. In the broad category of immunotherapy, cancer vaccines are of specific interest because they could provide synergy with other immune therapies by inducing or enhancing tumor-specific immune responses, and eventually, may be developed for both prophylactic and therapeutic use. One of the first therapeutic vaccines was developed in 1988 against melanoma and used homogenate from two different cell lines to vaccinate against a mix of uncharacterized antigens [1,2]. The drug Melacine was the result of this effort and successfully produced some anti-tumor activity in patients with metastatic melanoma [3]. This general strategy has been revised with the goal of producing more “personalized” vaccines containing tumor antigens from a patient’s own tumors, to generate a patient-specific immune response. These may contain either defined tumor antigens, usually discovered by sequencing and then delivered in a number of immunogenic preparations, or remain undefined and generated directly from resected tumors. This remains an exciting vein of research and multiple strategies of defining and cloning personalized tumor antigens, or producing personalized tumor vaccines are in development [1].

Although there are many systems for delivering specific tumor antigens, this review will focus on cytomegalovirus (CMV)-based vaccines. CMV is a ubiquitous beta herpes virus that normally produces an asymptomatic latent/persistent infection in immunocompetent hosts, although more severe disease can arise in transplant patients or during a congenital infection [4,5]. CMV establishes a systemic infection, ultimately infecting many cell types and most organs of the body [6,7,8]. In addition, CMV has the ability to superinfect hosts that are already CMV-positive [9,10]. Thus, it is possible that a CMV-based vaccine could be used to vaccinate either CMV-naïve or CMV-seropositive individuals, regardless of existing immune responses against any naturally acquired CMV. However, as discussed below, pre-existing immunity may influence the magnitude of immune responses targeting the vaccine antigens, while not precluding vaccination per se.

Most importantly, CMV causes an exceptionally large T cell response to CMV-encoded antigens. In particular, CD8^+^ T cell memory inflation—the accumulation of CMV-specific T cells over time—is the hallmark of a CMV infection, and results in CMV-specific T cell populations that average ~5–10% of all T cells in a healthy, CMV-infected host [11,12,13,14,15,16,17,18]. Significant work over the last several years has helped define this process of memory inflation. Data from the mouse model of murine (M)CMV infection showed that these T cell responses are antigen-dependent and directed against antigens that could be processed by the constitutive proteasome and presented by non-hematopoietic cells [19,20,21]. Evidence suggests that the human and murine viruses establish latency in myeloid cells [22,23,24,25,26,27,28,29,30], endothelial cells [31,32], and perhaps other cells of the body. Thus, the current model suggests that blips of viral reactivation from latency in non-hematopoietic cells repeatedly stimulate, and steadily expand, the virus-specific CD8^+^ T cell pool. Due to the persistence of the virus, these inflationary CD8^+^ T cells are maintained as effector or effector-memory T cells and not as resting central memory T cells [33]. Importantly however, such inflationary T cells do not show evidence of exhaustion, but rather maintain the ability to kill targets and release inflammatory cytokines [15,34,35,36]. Together, these features make CMV an attractive platform for vaccines and cancer therapy. 

To date, most work on a CMV-based vaccine has been conducted in the context of infectious diseases. Three of the most studied of these vaccines have used antigens from simian immunodeficiency virus (SIV) [37,38,39,40,41], *Zaire ebolavirus* (ZEBOV) [42,43,44], and *Mycobacterium tuberculosis* [45,46]. Individual studies of each vaccine in rhesus macaques using the rhesus (Rh)CMV platform showed a 30% protection rate from SIV, 75% protection rate from ZEBOV, and stronger responses than the current *M. tuberculosis* vaccine in 50–70% of animals [47]. Interestingly, the mechanisms of protection have varied. Protection against SIV, using multiple antigens, was correlated with T cell responses [38]. In mice vaccinated with MCMV encoding a known T cell epitope from the nuceloprotein of ZEBOV, protection likely depended on T cells [42,43]. However, Rhesus macaques vaccinated with Rhesus (Rh)CMV encoding the full-length glycoprotein from ZEBOV failed to promote ZEBOV-specific T cell responses and protection was instead associated with strong induction of antibodies [44]. It is still unclear exactly how protection against *M. tuberculosis* was achieved in mice or macaques, but the data may suggest that CMV established an innate immune environment that was resistant to *M. tuberculosis* infection [45,46]. Additional pre-clinical studies in mice have suggested a wide range of possible targets for CMV-based vaccination including an influenza A [48], sin nombre virus [49,50], and respiratory syncytial virus [51]. Thus, CMV-based vectors have received significant attention in the context of infectious diseases.

## 2. CMV As a Platform for Cancer Vaccines

The attributes that have led to the success of CMV as a vaccine against viruses and bacteria may also be relevant to the use of CMV as a cancer vaccine. The ability of CMV to engage both innate and adaptive immune responses in any patient regardless of prior exposure to the virus is a major benefit [9]. In addition, because CMV can establish a latent/persistent infection, a vaccine strain expressing a tumor antigen could continue to expand the patient’s circulating tumor-specific CD8^+^ T cell pool over time [11], potentially generating enough of a response to control tumor growth in a prophylactic or therapeutic setting and thwart metastatic growth at distant sites from the primary tumor. Furthermore, the virus typically causes no symptoms in an immunocompetent host during the course of a natural infection, meaning there may be fewer immediate safety concerns compared to other viral vectors [9]. Nevertheless, there are still significant safety concerns for the use of CMV, especially if vaccinating otherwise healthy people. Indeed, long-term CMV infection has been posited to contribute to a variety of inflammatory diseases such as inflammatory bowel disease [52] and cardiovascular disease [53,54,55], although the nature of the association is still unclear and not every clinical study has reached the same conclusions. Regardless, given any of these potential risks, strategies to attenuate the virus will be required. 

To date, MCMV has been successfully used in immunocompetent murine models as a vaccine encoding tumor antigens specific to prostate cancer [56], head and neck squamous cell carcinoma [57,58], and melanoma [59,60,61,62,63,64,65]. Understanding the mechanism of action by which each vaccine works will be key to developing CMV into a successful platform for cancer vaccines.

### 2.1. Prostate Cancer

The Jarvis lab generated a prostate cancer vaccine on a murine cytomegalovirus backbone that encodes human prostate-specific antigen (PSA) [56]. Using a single class I-restricted human PSA epitope (PSA_65–73_), they effectively generated an inflationary CD8^+^ T cell response against PSA in a humanized mouse model. The vaccine was tested in the TRAMP (transgenic adenocarcinoma of the mouse prostate) model and resulted in clearance of up to 87% of TRAMP-PSA tumors. This established the first proof-of-concept study for using CMV as a cancer vaccine.

### 2.2. Melanoma

#### 2.2.1. MCMV-TRP2

CMV has been most studied as a cancer vaccine in melanoma with multiple labs finding efficacy in separate melanoma models. The Hill lab was first to produce a CMV vaccine for melanoma and encoded the unmodified melanoma antigen TRP2 into the viral genome [62]. A single dose of MCMV-TRP2 was sufficient to cause rejection of the aggressive B16-F10 melanomas in a prophylactic setting and also delayed tumor growth in a therapeutic model. Surprisingly, however, the efficacy of the vaccine was not dependent on a CD8^+^ T cell response. In fact, MCMV-TRP2 failed to develop substantial numbers of tumor-specific CD8^+^ T cells and depletion of T cells had no impact on tumor protection. Rather, in this model, tumor-specific antibodies were critical for protection against B16-F10 tumors. Subsequent work from the Arens and Verbeek labs showed that protection in this model required expression of FcγRI on macrophages [65].

#### 2.2.2. MCMV-gp100

A second set of CMV-based melanoma vaccines have been produced containing the gp100 antigen. Interestingly, the Khanna lab demonstrated that the native (self) gp100 antigen was not sufficient to promote tumor-specific CD8^+^ T cells [63]. However, a modified gp100 antigen, mutated to increase the peptide affinity for MHC-I, induced robust CD8^+^ T cell responses. This vaccine had efficacy in both prophylactic and therapeutic settings in a B16-F10 model when the tumors were seeded intravenously to establish nodules in the lungs, and as expected, this vaccine was dependent of CD8^+^ T cells. Disappointingly, when the B16-F10 tumors were implanted into the skin, the vaccine was barely protective in a therapeutic setting, a result also seen in our own lab using an independently-generated MCMV-gp100 vaccine [59,64]. However, the Khanna lab demonstrated that adoptive T cell therapy could boost the number of T cells and provide increased efficacy [59].

### 2.3. Head and Neck Squamous Cell Carcinoma

One challenge to developing a tumor vaccine is that the ideal target antigens may be mutations that generate novel T cell epitopes in the tumor, which will differ between individuals. One exception may be tumors driven by human papilloma virus (HPV), which transforms epithelial cells by uncontrolled expression of the viral E6 and E7 proteins. The Cicin-Sain and Arens labs have developed MCMV-based vaccines expressing either a single epitope of HPV E7 (E7_49–57_) or the full-length E6 and E7 proteins [57,58]. This work demonstrated that the single epitope, especially if encoded at the C-terminal end of the ie2 protein to promote peptide processing, was more effective than the full-length protein at generating robust effector-memory CD8^+^ T cell responses and protecting against TC-1 squamous cell carcinomas (HPV E6 and E7 positive) [57]. The Arens lab further characterized the optimal immune responses for tumor protection, and showed that E7-specific CD8^+^ T cells that exceeded 0.3% of the total CD8^+^T cell population in the blood provided full protection against tumor challenge. Thus, this study identified a threshold in the magnitude of the tumor-specific response, which was needed for full protection, which agreed with the work from the Khanna lab in the skin-melanoma model. Interestingly, the Arens lab further showed that prior exposure to MCMV did not prevent the formation of new tumor-specific CD8^+^ T cell responses elicited by subsequent vaccination. These data agree with prior work showing that CMV can super-infect previously infected hosts, leading to new immune responses. However, their data suggest that the magnitude of the immune response to the primary infection modulated the magnitude of the immune response to vaccination. Thus, in animals with a robust response to the primary MCMV infection, the magnitude of the tumor-specific response after vaccination was reduced, which limited protection against TC-1 challenge, presumably as a result of vector-specific immune responses [58]. Thus, prior CMV infection may reduce the quantity of vaccine-driven T cells, which could reduce the impact of vaccination.

## 3. Attenuating the CMV Vector to Increase Safety

Although CMV causes an asymptomatic infection in healthy, immune-competent people, the virus will likely need to be attenuated to further increase the safety profile. The Koszinowski lab demonstrated that a spread-defective MCMV lacking the essential M94 gene, which was unable to spread beyond the first cells infected and, therefore, was safe for mice lacking the Type I IFN receptor, persisted in mice for at least 1 year in vivo [66,67]. Similarly, our own previous work demonstrated that a spread-defective MCMV lacking the essential glycoprotein L (ΔgL), was safe for mice with severe combined immune deficiency (SCID), but nevertheless persisted in vivo and promoted CD8^+^ T cell memory inflation in wild-type mice [68,69]. Thus, spread defective CMV vectors may be an ideal way to improve the safety profile of this vector. Importantly, the Hill lab demonstrated that the ΔgL-MCMV-TRP2 vaccine was equally protective against B16-F10 melanomas in the model that depended on tumor-specific antibodies [62]. The Arens lab also attenuated CMV by attaching the FKBP-degradation domain to M79, which restricts viral replication to a single cycle but was still able to generate sufficient numbers of CD8^+^ T cells specific to the vaccine antigen HPV E7 [58]. Interestingly, an alternative strategy has been proposed by Jonjic and colleagues in which the MCMV vaccine vector encoded the RAE-1γ molecule, a ligand for the NK and CD8^+^ T cell co-stimulatory receptor, NKG2D. This MCMV construct was severely attenuated in vivo as a result of robust NK cell-mediated control and was safe in immunocompromised animals. Remarkably, when expressing the SIINFEKL epitope from ovalbumin, this vector enhanced the anti-tumor immune response against B16-OVA and EG7 tumor cells (both Ova-expressing) compared to a vaccine lacking RAE-1γ [61]. Thus, there may be ways to enhance the vaccine efficacy while increasing its safety profile.

## 4. The Prospect for CMV-Based Vaccines for Cancer

Together, these pre-clinical data suggest that CMV may be effective at inducing anti-tumor immunity in certain settings. However, these studies also raise several significant caveats. First, the data suggest that the persistence of the CMV vector, and the maintenance of effector-memory T cells, may not be enough to provide protection. Rather, vaccine efficacy may ultimately depend on achieving a sufficiently large number of tumor-specific CD8^+^ T cells [58]. Reaching this threshold will clearly depend on the quality of the antigen and the presence of pre-existing immunity and it is unclear whether this threshold can be reached with natural tumor antigens in people that are likely already CMV-positive. Second, choosing the right antigen will be a challenge. The use of full-length antigens has been less effective at generating tumor-specific CD8^+^ T cell responses, and therefore, less effective at providing protection than encoding specific peptides in the CMV backbone [57,62,63]. Moreover, the native TRP2 and gp100 antigens mostly failed to induce CD8^+^ T cell responses, possibly suggesting that modified antigens, or peptide antigens containing tumor-associated mutations, will be needed to drive significant tumor-specific responses from a CMV backbone [62,63]. Given the diversity of MHC in people, these issues will make it very difficult to produce an “off-the-shelf” vaccine that can be used in a larger population. Third, the tumor site may dictate the efficacy of the vaccine. Melanoma nodules growing in the lungs after intravenous injection were far better controlled than a mass growing in the skin [59,63]. This may be due to the fact that most inflationary T cells are circulating in the blood [16] and, therefore, have easy access to the lung nodules, or it could indicate that a different threshold of T cells is required to control tumors in different sites. In either case, vaccine strategies may need to be designed and modified for different situations. Finally, CMV itself has been proposed to promote or exacerbate tumor growth [70,71]. Thus, a CMV-based cancer vaccine could conceivably be counter-productive. Interestingly, however, a few studies have also suggested that CMV might promote endogenous anti-tumor immune responses in some settings. It is, therefore, imperative to determine how CMV alters the tumor environment.

## 5. Potential Off-Target Effects of CMV in the Tumor

### 5.1. CMV as a Cancer Promoter

Studies dating back to the early 1970′s from the Rapp lab have shown the potential of CMV to transform embryonic fibroblasts and embryonic lung cells in vitro. These infected cells caused tumor formation when injected into immune deficient mice and hamsters and generated a CMV-specific antibody response in the hamsters as well [72,73,74]. However, it was also shown that the transformed cells lost expression of CMV antigens over time, indicating that there was another factor that allowed these cells to stay transformed [75]. Through multiple studies, a theory emerged that CMV may not be directly oncogenic, but that it may be able to alter an already transformed cell in a way that makes it more malignant. During a normal infection, expression of CMV regulatory proteins can promote cell cycle arrest while simultaneously forcing the cell into S phase to ensure viral genome replication. During this process, HCMV strictly regulates expression of pRB and p53 [71,76,77]. However, in tumor lines lacking pRb and p53, HCMV can no longer induce cell cycle arrest and the cells will continue to divide. Therefore, HCMV can drive both cell proliferation or cell cycle arrest, depending on the state of the infected cell [78]. Additionally, HCMV is known to decrease apoptosis and increase cellular migration and adhesion molecule expression, all of which can promote aggressive properties of already transformed cells [71]. Thus, if the tumor cells are directly infected, HCMV might enhance tumor progression. Additionally, indirect effects of HCMV have been proposed. For example, HCMV mediated induction of IL-6 may also promote angiogenesis [79], effects that would be independent of direct tumor cell infection. Thus, CMV in the tumor stroma may also promote tumor growth.

Some clinical work supports the hypothesis that CMV may promote tumor growth, although this claim is still contested [80,81,82,83,84,85,86,87,88,89,90,91,92]. Tumor samples from ovarian cancer [88], non-melanoma skin cancer [89], colorectal adenocarcinoma [90], rhabdomyosarcoma [91], and glioblastoma [92] have all tested positive for CMV in the tumor. Additionally, some studies report that poor outcomes in ovarian cancer may correlate with the amount of CMV detected in the tumor [88]. However, other studies have failed to find CMV within tumors, even when using very sensitive genome sequencing approaches [80,81,82,83,84,85,86,87]. Moreover, it is not yet clear whether CMV, if it is detected in tumors, is promoting or exacerbating tumor growth, or simply present within those tumors and more active within more aggressive tumors. Perhaps the most direct evidence for a role of CMV in tumor growth comes from studies in which patients with glioblastoma and CMV were treated with the antiviral drugs, ganciclovir and cidofovir, and exhibited slower tumor growth [93,94,95]. However, off-target effects of ganciclovir, which has been shown to potently inhibit neuroinflammation [96], have not been excluded. Moreover, as ganciclovir should kill cells with replicating virus, these data cannot exclude the possibility that increasing cell death in the tumor is associated with improved outcomes, even if CMV played no role in tumor growth. Thus, more direct experimental evidence will be needed to explore the interaction of CMV with the tumor environment.

Some recent experimental studies in mice have supported the possibility that CMV can promote tumor growth. First, work from the Chiocca lab showed that MCMV promoted rhabdomycosarcoma formation in mice lacking one copy of p53 [91]. Interestingly, mice had to be infected perinatally for tumors to develop, which results in altered viral distribution [97]. Additionally, all tumors that developed in this model had lost the second copy of p53. Likewise, a second study indicated that perinatal MCMV infection reduced survival in a spontaneous genetic model of glioma in which the mice lacked one copy of Nf1 in the brain, and one copy of p53 in all cells [98]. In this model, the authors identified virus-induced activation of STAT3 as a potential mediator of tumor progression. More recently, the Lawler and Chiocca groups provided a potential mechanism by which MCMV might promote tumor growth through increased blood flow and angiogenesis in an orthotopic glioblastoma model [99]. Finally, recent experimental studies in a murine model of spontaneous breast cancer growth with lung metastasis showed that neither active nor latent MCMV had an impact on primary tumor growth, but that an increase in metastasis was found in animals with latent MCMV [100]. This study further supports the work from the Lawler and Chiocca groups as it identified an increase in angiogenesis to the tumors after MCMV infection. Together, these data suggest that under certain experimental conditions, the presence of CMV in the host may contribute to tumor growth or progression.

### 5.2. Potential Anti-Tumor Effects of WT-CMV

The first indication that CMV infection may not always promote tumor growth came from the Reddehase lab, which showed that MCMV infection in mice delayed the growth of lymphoma cells in a bone marrow transplant model [101,102]. In this setting, the lymphoma cells were not infected by CMV and T cells were not needed. Instead, MCMV seemed to induce an “apoptotic crisis” in the lymphoma cells through local production of IL-15 among other, still undefined mechanisms [103].

In a different model, we showed that MCMV could delay the growth of well-established B16 melanomas when injected intratumorally (IT). This effect was dependent on the endogenous anti-tumor immune response, and in particular, CD8^+^ T cells, but did not depend on encoding tumor antigens in the viral genome [64]. Excitingly, IT-MCMV synergized with a PD-L1 blockade to clear ~70% of these melanomas, a tumor that is otherwise almost completely resistant to PD-1 therapy [64,104]. Additional and ongoing work has suggested that MCMV was not acting as an oncolytic virus (i.e., directly infecting and killing tumor cells) in this model, but rather was infecting and modifying tumor-associated macrophages (TAMs, manuscript submitted [105]). In fact, the major cells infected in this model were TAMs [64].

Although CMV infects macrophages in a natural infection setting [23], it was interesting that TAMs were primarily infected in this model because they are important modulators of the immune response within a tumor and can have both pro-tumor and anti-tumor activity. Macrophages are predominantly differentiated towards an anti-inflammatory (M2-like) state in most solid tumors and can contribute to the growth of the lesion in this state [106,107,108,109,110,111]. For this reason, significant effort has been aimed at depleting macrophages from tumors in a clinical setting [112]. However, it has become apparent that if TAMs are converted to an inflammatory (M1-like) state, they can promote anti-tumor immune responses [113,114]. In fact, recent evidence suggests that activating TAMs may lead to anti-tumor T cell responses [109].

HCMV is well-known to polarize human peripheral blood monocytes and macrophages toward an inflammatory state [115,116,117]. Likewise, we have found that MCMV drives M2-like macrophages into an M1-like state. Moreover, depletion of phagocytic cells during IT-MCMV injections of B16 tumors prevented MCMV from delaying tumor growth [105]. Most interestingly, it was expression of the viral chemokine MCK-2, which is involved in recruiting monocytes to the site of infection, that was needed to recruit TAMs to the B16 tumors [105,118,119]. Thus, MCMV may delay tumor growth in this model by recruiting and polarizing TAMs to promote anti-tumor immunity. These data suggest that CMV could potentially be used as an adjuvant to increase immune infiltration into immunologically “cold” tumors and to activate local myeloid cells to engage adaptive immune responses.

### 5.3. Active vs. Inactive CMV: A Hypothesis for the Opposite Effects of CMV in Different Settings

Accumulating evidence suggests that CMV may alter tumor cells to decrease apoptosis, while increasing proliferation, adhesion, tumor cell and stromal cell migration, and angiogenesis [71] (Figure 1). However, while there is strong evidence that CMV is present in a variety of tumors, it is still unclear what sort of viral gene expression occurs in these tumors. Thus, it is largely unknown whether the virus is replicating or in an undefined state of latency or semi-reactivation with limited gene expression. From our reading of the literature, there is little evidence to suggest that CMV is actively replicating in patient tumors.

In contrast, evidence from our lab and the Reddehase lab [101,102,103], suggest that active CMV replication may be associated with inflammatory responses that are inhibitory to tumor growth and/or can engage adaptive immune responses. Thus, we propose a model in which the state of CMV in the tumor will be a crucial factor in determining whether the infection promotes tumor growth and progression, or whether the virus tips the scales toward tumor delay and immune responses (Figure 1). Latency or a state of semi-reactivation may not trigger the innate inflammatory response of the host, nor engage anti-viral or anti-tumor T cell and NK cell responses. In contrast, active viral infection should drive strong interferon responses, recruit activated inflammatory myeloid cells, and promote adaptive immune responses in the tumor. Thus, injections of CMV into a tumor to generate an active infection may change the relationship between the tumor, CMV, and the immune response.

Testing this model will depend on more fully defining the viral gene expression and the cellular response to CMV (e.g., Is Type I IFN increased? Have TAMs, T cells, or NK cells accumulated or become activated as a result of CMV infection?). In addition, we will need to expand on the experimental models that have been used and test conditions when CMV is either latent or actively replicating. Thus, it will be exciting and informative to know whether the glioma growth and progression that is accelerated by latent MCMV [99] can be reversed by infecting the tumor with new virus to achieve active replication. If so, such a result could suggest that therapies designed to drive full reactivation of a patient’s endogenous CMV might promote inflammation and anti-tumor immune responses, switching the relationship between CMV and the tumor. Likewise, it will be important to test whether the presence of latent MCMV can accelerate the growth of other tumors, particularly mouse melanoma in the skin [64,105] but also tumors implanted in other sites or derived from other tissues.

It will also be important to determine whether the tumor cell type or the type of cell infected by CMV in the tumor environment plays a role in tumor suppression or promotion. For this, future work should focus on understanding virus localization in both experimental settings and within human tumors. Additionally, in models in which tumor cells are not the primary infected cell type (e.g., B16-F0, lymphoma) future work could address whether adapting the virus to growth in the tumor cells has any impact on anti-tumor immune responses. Overall, this may help to inform the types of tumors that would be most impacted by a therapeutic CMV vaccine.

Finally, we think that it will be important to explore whether the lab-adapted and vaccine strains of CMV used in experimental models accurately reflect the behavior of wild strains of CMV. It is conceivable that the wild, circulating strains of CMV will be much less inflammatory, and much more readily associated with tumor progression than lab-adapted strains that are generally selected for their ability to produce high viral titers in culture. Several wild strains of MCMV have been isolated and characterized by the Redwood group [120,121] and these may be very useful in determining whether the effect of MCMV on the tumor is partially due to the high passage strains that have been studied to date.

CMV has undoubtedly shown promise as a vaccine vector for infectious diseases. However, when thinking about CMV as a cancer vaccine, more consideration must be given to the impact of CMV on the tumor itself. Conflicting literature suggests that CMV can both promote and inhibit the growth of a tumor in different contexts. Thus, developing cancer therapies based on the CMV backbone requires detailed study of the interplay between the virus, the tumor, and the tumor microenvironment.

## Figures and Tables

**Figure 1 vaccines-07-00062-f001:**
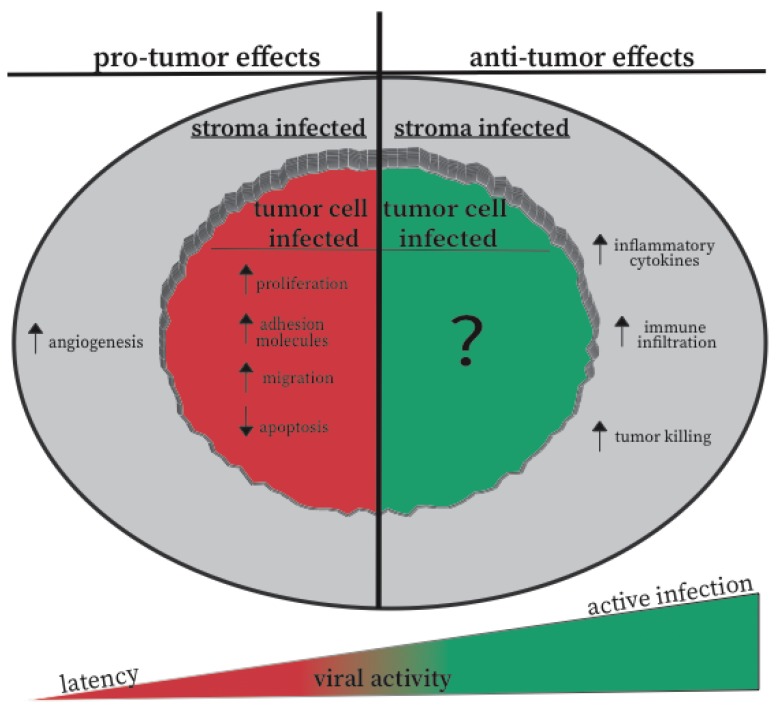
Effects of CMV on the tumor cells and microenvironment. The schematic summarizes the experimental evidence for both CMV-mediated tumor promotion and CMV-mediated tumor regression. These effects are classified as to whether the effect can be explained by CMV infection of the tumor cells, or by infection of other cells in the tumor microenvironment. Our hypothesis for how viral activity contributes to these outcomes is also displayed by the wedge-shape on the bottom of the image.

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
