# Peer review of "From Vaccine Vector to Oncomodulation: Understanding the Complex Interplay between CMV and Cancer"

_vaccines, 2019, doi:10.3390/vaccines7030062_

Round 1
Reviewer 1 Report
The review by Wilski describes the pros and cons of CMV as a vaccine vector for cancer. Overall the review is well written and describes the majority of the current knowledge. Besides the potential impact of CMV-based vaccines also potential detrimental effects of CMV infection on tumor progression are discussed.
A few suggestions and comments:
-In the abstract I miss the hypothesis of the review (5.3).
-2.2.1. Here the paper by Benonisson H et al (Oncotarget, 2018) on the mechanisms of MCMV-TRP2 is missing.
-3. Attenuating the CMV vector: Work with the spread-defective M79-FKBP vectors is missing (e.g. Beyranvand Nejad, 2019).
- The authors discuss the pro- and anti-tumorigenic role of CMV. However, there may be a major difference in using CMV-based vaccines vectors versus natural CMV infection in this respect. This difference may be better discussed for clarity.
-Future directions? Based on the hypothesis: do the authors suggest to develop CMV-based vaccines targeting immune cells and (not) tumor cells and/or stroma? Is intra-tumoral the preferred vaccination route? Should the vaccine be attenuated (or not) but still provide sufficient inflammation? Some directions and thoughts could be provided.
Author Response
Thank you very much for the feedback. All comments have been addressed and are listed as follows:
In the abstract I miss the hypothesis of the review (5.3).
We have added a formal hypothesis to the abstract. Please see lines 25-27.
2.2.1. Here the paper by Benonisson H et al (Oncotarget, 2018) on the mechanisms of MCMV-TRP2 is missing.
Failure to include this reference was an oversight. We have added it to the discussion of CMV vaccines for melanoma. Please lines 95-96 and 114-115.
3. Attenuating the CMV vector: Work with the spread-defective M79-FKBP vectors is missing (e.g. Beyranvand Nejad, 2019).
This too was an oversight that we have now corrected on lines 160-162
- The authors discuss the pro- and anti-tumorigenic role of CMV. However, there may be a major difference in using CMV-based vaccines vectors versus natural CMV infection in this respect. This difference may be better discussed for clarity.
We have expanded on this discussion on lines 312-319
-Future directions? Based on the hypothesis: do the authors suggest to develop CMV-based vaccines targeting immune cells and (not) tumor cells and/or stroma? Is intra-tumoral the preferred vaccination route? Should the vaccine be attenuated (or not) but still provide sufficient inflammation? Some directions and thoughts could be provided.
We have expanded on this discussion on lines 297-311 and provided some specific future directions we would like to see.
Reviewer 2 Report
The review addresses equally well both the advantages and concerns of the platform and overall is well written making it a very approachable review
Two minor comments and questions for clarification:
Line 109-111 – Is this sentence is conveying the correct meaning? The implication is that tumour protection is mediated by MCMV-specific antibodies, but they mean TRP2-specific antibodies induced by the MCMV-TRP2 vaccine? A similar confusion for the reader is apparent when talking about T cells. Induction of MCMV-specific T cells is not relevant to tumour control, but TRP2-specific T cells. MCMV-specific immune responses would only be relevant if you want to talk about immunodominance of immune responses against the MCMV backbone and the heterologous antigens delivered by the vaccine, which is not addressed in the review and could be an interesting topic to be added.
Line 136-140 – This sentence is confusing. The authors state that primary infection didn’t hindered overall size of the response (what do you mean by size? Percentage of antigen-specific T cells?) but reduced efficacy and concluded that large numbers of T cells may be necessary for protection. Based on what was written, since “size” was the same, but response was worse, one cannot conclude that large numbers are an important thing for protection. It would be helpful to clarify this.
Author Response
Thank you very much for your feedback. Your comments have all been addressed as follows:
Line 109-111 – Is this sentence is conveying the correct meaning? The implication is that tumour protection is mediated by MCMV-specific antibodies, but they mean TRP2-specific antibodies induced by the MCMV-TRP2 vaccine? A similar confusion for the reader is apparent when talking about T cells. Induction of MCMV-specific T cells is not relevant to tumour control, but TRP2-specific T cells. MCMV-specific immune responses would only be relevant if you want to talk about immunodominance of immune responses against the MCMV backbone and the heterologous antigens delivered by the vaccine, which is not addressed in the review and could be an interesting topic to be added.
Our goal was to discuss the attributes of a CMV-based vaccine strain encoding tumor antigens in the viral backbone and driving tumor-specific responses along with virus-specific responses. We did not mean to state that MCMV-specific responses were contributing. This has now been clarified in the text, lines 88-91, 94-95, and 112.
Line 136-140 – This sentence is confusing. The authors state that primary infection didn’t hindered overall size of the response (what do you mean by size? Percentage of antigen-specific T cells?) but reduced efficacy and concluded that large numbers of T cells may be necessary for protection. Based on what was written, since “size” was the same, but response was worse, one cannot conclude that large numbers are an important thing for protection. It would be helpful to clarify this.
We meant to discuss the possibility that prior immunity to CMV could reduce the size of the response to vaccination with a CMV vector. We have attempted to clarify this issue on lines 138-148.